# Mitochondria Play Essential Roles in Intracellular Protection against Oxidative Stress—Which Molecules among the ROS Generated in the Mitochondria Can Escape the Mitochondria and Contribute to Signal Activation in Cytosol?

**DOI:** 10.3390/biom14010128

**Published:** 2024-01-19

**Authors:** Daisuke Masuda, Ikuo Nakanishi, Kei Ohkubo, Hiromu Ito, Ken-ichiro Matsumoto, Hiroshi Ichikawa, Moragot Chatatikun, Wiyada Kwanhian Klangbud, Manas Kotepui, Motoki Imai, Fumitaka Kawakami, Makoto Kubo, Hirofumi Matsui, Jitbanjong Tangpong, Takafumi Ichikawa, Toshihiko Ozawa, Hsiu-Chuan Yen, Daret K. St Clair, Hiroko P. Indo, Hideyuki J. Majima

**Affiliations:** 1Department of Space Environmental Medicine, Kagoshima University Graduate School of Medical and Dental Sciences, Kagoshima 890-8544, Kagoshima, Japan; masudadaisuke@yahoo.co.jp; 2Utilization & Engineering Department, Japan Manned Space Systems Corporation, 2-1-6 Tsukuba, Tsukuba 305-0047, Ibaraki, Japan; 3Quantum RedOx Chemistry Team, Institute for Quantum Life Science (iQLS), Quantum Life and Medical Science Directorate (QLMS), National Institutes for Quantum Science and Technology (QST), 4-9-1 Anagawa, Inage-ku, Chiba 263-8555, Japan; ito.hiromu@qst.go.jp; 4Institute for Advanced Co-Creation Studies, Open and Transdisciplinary Research Initiatives, Osaka University, Suita 565-0871, Japan; ohkubo@irdd.osaka-u.ac.jp; 5Department of Maxillofacial Radiology, Field of Oncology, Graduate School of Medical and Dental Sciences, Kagoshima University, Kagoshima 890-8544, Kagoshima, Japan; 6Quantitative RedOx Sensing Group, Department of Radiation Regulatory Science Research, Institute for Radiological Science (NIRS), Quantum Life and Medical Science Directorate (QLMS), National Institutes for Quantum Science and Technology (QST), 4-9-1 Anagawa, Inage-ku, Chiba 263-8555, Japan; matsumoto.kenichiro@qst.go.jp; 7Department of Medical Life Systems, Graduate School of Life and Medical Sciences, Doshisha University, Kyoto 610-0394, Kyoto, Japan; hichikaw@mail.doshisha.ac.jp; 8School of Allied Health Sciences, Walailak University, Thasala, Nakhon Si Thammarat 80161, Thailand; moragot.ch@wu.ac.th (M.C.); kwiyada@wu.ac.th (W.K.K.); manas.ko@wu.ac.th (M.K.); rjitbanj@wu.ac.th (J.T.); 9Center of Excellence Research for Melioidosis and Microorganisms, Walailak University, Thasala, Nakhon Si Thammarat 80161, Thailand; 10Regenerative Medicine and Cell Design Research Facility, School of Allied Health Sciences, Kitasato University, 1-15-1 Kitasato, Sagamihara 252-0373, Kanagawa, Japan; imai-m@kitasato-u.ac.jp (M.I.); kawakami@kitasato-u.ac.jp (F.K.); kuboma@kitasato-u.ac.jp (M.K.); t.ichika@kitasato-u.ac.jp (T.I.); 11Department of Molecular Diagnostics, School of Allied Health Sciences, Kitasato University, 1-15-1 Kitasato, Sagamihara 252-0373, Kanagawa, Japan; 12Department of Regulation Biochemistry, Kitasato University Graduate School of Medical Sciences, 1-15-1 Kitasato, Sagamihara 252-0373, Kanagawa, Japan; 13Department of Health Administration, School of Allied Health Sciences, Kitasato University, 1-15-1 Kitasato, Sagamihara 252-0373, Kanagawa, Japan; 14Division of Microbiology, Kitasato University School of Allied Health Sciences, 1-15-1 Kitasato, Minami-ku, Sagamihara 252-0373, Kanagawa, Japan; 15Department of Environmental Microbiology, Graduate School of Medical Sciences, Kitasato University, 1-15-1 Kitasato, Minami-ku, Sagamihara 252-0373, Kanagawa, Japan; 16Division of Gastroenterology, Graduate School of Comprehensive Human Science, University of Tsukuba, Tsukuba 305-8575, Ibaraki, Japan; hmatsui@md.tsukuba.ac.jp; 17Research Excellence Center for Innovation and Health Products (RECIHP), School of Allied Health Sciences, Walailak University, Thasala, Nakhon Si Thammarat 80160, Thailand; 18Nihon Pharmaceutical University, 10281 Komuro, Ina-machi, Kitaadachi-gun, Saitama 362-0806, Saitama, Japan; ozawa@rugbygoods.com; 19Department of Medical Biotechnology and Laboratory Science, College of Medicine, Chang Gung University, Taoyuan 33302, Taiwan; yen@mail.cgu.edu.tw; 20Department of Nephrology, Chang Gung Memorial Hospital at Linkou, Taoyuan 33305, Taiwan; 21Department of Toxicology and Cancer Biology, University of Kentucky College of Medicine, Lexington, KY 40536, USA; daret.stclair@uky.edu

**Keywords:** mitochondria, reactive oxygen species, dipole moment, cell signaling, signal transduction, Nrf2/Keap1

## Abstract

Questions about which reactive oxygen species (ROS) or reactive nitrogen species (RNS) can escape from the mitochondria and activate signals must be addressed. In this study, two parameters, the calculated dipole moment (debye, D) and permeability coefficient (Pm) (cm s^−1^), are listed for hydrogen peroxide (H_2_O_2_), hydroxyl radical (•OH), superoxide (O_2_^•−^), hydroperoxyl radical (HO_2_•), nitric oxide (•NO), nitrogen dioxide (•NO_2_), peroxynitrite (ONOO^−^), and peroxynitrous acid (ONOOH) in comparison to those for water (H_2_O). O_2_^•−^ is generated from the mitochondrial electron transport chain (ETC), and several other ROS and RNS can be generated subsequently. The candidates which pass through the mitochondrial membrane include ROS with a small number of dipoles, i.e., H_2_O_2_, HO_2_•, ONOOH, •OH, and •NO. The results show that the dipole moment of •NO_2_ is 0.35 D, indicating permeability; however, •NO_2_ can be eliminated quickly. The dipole moments of •OH (1.67 D) and ONOOH (1.77 D) indicate that they might be permeable. This study also suggests that the mitochondria play a central role in protecting against further oxidative stress in cells. The amounts, the long half-life, the diffusion distance, the Pm, the one-electron reduction potential, the p*K*_a_, and the rate constants for the reaction with ascorbate and glutathione are listed for various ROS/RNS, •OH, singlet oxygen (^1^O_2_), H_2_O_2_, O_2_^•−^, HO_2_•, •NO, •NO_2_, ONOO^−^, and ONOOH, and compared with those for H_2_O and oxygen (O_2_). Molecules with negative electrical charges cannot directly diffuse through the phospholipid bilayer of the mitochondrial membranes. Short-lived molecules, such as •OH, would be difficult to contribute to intracellular signaling. Finally, HO_2_• and ONOOH were selected as candidates for the ROS/RNS that pass through the mitochondrial membrane.

## 1. Introduction

Reactive oxygen species (ROS) and reactive nitrogen species (RNS) consist of both radical and nonradical molecules and are reactive species that have different degrees of oxidizing potential in biological systems [1]. Many chronic diseases, such as cancer, alcoholic liver disease, Crohn’s disease, rheumatoid arthritis, diabetes, muscular dystrophy, cystic fibrosis, septic shock, premature babies, atherosclerosis, infertility, cataracts, aging, hepatitis, ARDS, ischemia, neuronal degeneration, etc., are recognized as oxidative-stress-related diseases (OSDs) [2]. A major source of ROS in cells is the mitochondria [3]. The electron transport chain (ETC) consists of Complexes I, II, III, and IV. Oxidative phosphorylation is the process of the coupling between the ETC and ATP production in Complex V. Mitochondrial DNA (mtDNA) encodes 13 proteins inside the mitochondrial matrix, and those proteins are parts of Complexes I, III, IV, and V. [4]. Overall, 2~3% of electrons leak from the ETC and oxygen captures them, resulting in the production of superoxide anions (O_2_^•−^). It is well known that mitochondria are the major site of ATP production, but they also produce O_2_^•−^, which mainly leaks from Complexes I and III [2]. Impairment of the ETC caused by chemicals or mtDNA damage can cause an increase in the generation of O_2_^•−^ and subsequent ROS [3]. These impairments are closely related to the cause of OSDs [4,5]. Hydroperoxyl radical (HO_2_•) is the protonated form of O_2_^•−^, but whether its amount could be affected by the pH gradient across the mitochondrial inner membrane is uncertain [6]. There is evidence of nitic oxide (•NO) formation in the mitochondria, although whether mitochondrial nitric oxide synthase (NOS) exists is still controversial [7]. Singlet oxygen (^1^O_2_) can be generated endogenously through different mechanisms [8], but its formation in the mitochondria has only been addressed in one study [9].

In mammalian cells, there are three superoxide dismutase (SOD) isoenzymes: copper–zinc SOD (CuZnSOD), or SOD1 [10]; manganese SOD (MnSOD), or SOD2 [11]; and extracellular SOD (ECSOD), or SOD3 [12]. SOD catalyzes the dismutation of two superoxide radicals into hydrogen peroxide and oxygen. MnSOD is an enzyme localized in the mitochondrial matrix. Okado-Matsumoto and Fridovich showed that CuZnSOD is localized in the intermembrane space of the mitochondria [13]. It has been recognized that increases in the generation of ROS from the mitochondria can cause lipid oxidation and apoptosis. MnSOD could protect against these processes [14].

How do antioxidant systems, which are intracellular defense systems, work? MnSOD generates one hydrogen peroxide (H_2_O_2_) from two superoxide radicals (O_2_^•−^). MnSOD may also reduce the formation of hydroxyl radicals (•OH) from superoxide (O_2_^•−^) and hydrogen peroxide (H_2_O_2_) through the Haber–Weiss reaction under the catalysis of iron ions [15,16,17]. However, H_2_O_2_ from MnSOD could be quickly detoxified by mitochondrial glutathione peroxidase (mtGPx) by reducing it to water [14,18]. This reaction could be accompanied by glutathione, of which the level for most cells is ~5 mM, an excess amount for the reaction [14,18]. Furthermore, GPx4 knockout (KO) is known to cause acute renal failure and death [19,20], suggesting that GPx4 plays an essential role as an antioxidant in mitochondria. Due to the emergence of the role of nitric oxide (•NO) in OSDs, reactive nitrogen cascades are sometimes included in reactive oxygen cascades. O_2_^•−^ and •NO can be easily bound and produce peroxynitrite (ONOO^–^) with *k* = 5 × 10^9^ M^−1^ s^−1^; however, in the opposite reaction, *k* = 0.023 s^−1^ [21]. ONOOH produces •NO_2_ and •OH with *k* = 0.35 s^−1^, indicating that the decomposition of ONOO^–^ and ONOOH is not straightforward [21]. Kissner et al. (2003) suggested that, regarding peroxynitrite formation under physiological conditions, when 10 nM •NO and 10 µM SOD, ONOO^–^ formation/O_2_^•−^ dismutation is 1/125, while with 2 µM •NO and 2 µM SOD, ONOO^–^ formation/ O_2_^•−^ dismutation is 8/1 [22], suggesting that ONOO^–^ formation is dependent on intracellular •NO concentration. 

Mitochondrial ROS (mtROS) might be related to an increase in signal transduction and may control anti-oxidative-stress-related molecular defense mechanisms. Redox states could thus represent essential pathways to maintain homeostasis. The importance of this subject, the mitochondrial ROS come out from mitochondria and initiate the signal transduction inside cells, has been hypothesized by many researchers [23,24,25,26,27,28,29,30,31,32,33]. The role of mitochondrial ROS in initiating signal transductions in the cell cytosol has been the subject of discussion [34]. Indo et al. showed that manganese superoxide dismutase (MnSOD) transfection decreases the expression levels of GATA 1, 3, 4, and 5, which are nuclear factor kappa-light-chain-enhancer of activated B cells (NF-κB) regulating genes [34]. The results showed that MnSOD transfected cells revealed a decrease in expression compared to those in the control. We previously demonstrated that mtROS causes intracellular signaling, and we published a paper entitled “Evidence of Nrf2/Keap1 Signaling Regulation by Mitochondrial-Generated Oxygen Species in RGK1 cells” in a Special Issue of *Biomolecules* entitled “The Physiological and Pathological New Function of Mitochondrial ROS and Intraorganellar Cross-Talks” in 2023 (https://www.mdpi.com/journal/biomolecules/special_issues/0XTJ2MAYET, accessed on 7 November 2023) [35]. They transfected MnSOD gene-contained vectors in a gastric mucosal tumorized cell line, RGK1 cells. They examined the expression levels of NF-E2-related factor 2 (Nrf2), Kelch-like ECH-associated protein1 (Keap1), heme oxygenase-1 (HO-1) and 2, MnSOD, glutamate-cysteine ligase (GCL), glutathione S-transferase (GST), and NAD(P)H Quinone oxidoreductase 1 (NQO1), which are all Nrf2-Keap1 regulating gens. The results of immunocytochemistry staining showed a decrease in those expressions in the MnSOD transfected RGK1 cells compared to those in the control. The transfected MnSOD gene should decrease the mitochondrial ROS levels, so after MnSOD transfection, all decreased expression was shown, suggesting mtROS levels control the levels of Nrf2-Keap1 regulating genes. However, the question of which ROS go out from mitochondria and contribute to intracellular signaling remains unclear. 

The plasma membrane consists of both lipids and proteins. The fundamental structure of the membrane is the phospholipid bilayer, which forms a stable barrier between two aqueous compartments [36]. Most biologically important solutes require protein carriers to cross cell membranes, via a process of either passive or active transport. Active transport requires the cell to expend energy to move the materials, while passive transport can be performed without using cellular energy [37]. Certain substances easily pass through the membrane through passive diffusion, such as O_2_ and CO_2_, along with small relatively hydrophobic molecules, fatty acids, and alcohols [37]. Mitochondria possess double membranes, and the inner membrane contains cardiolipin. Cardiolipin is not the main lipid that forms a phospholipid bilayer but fulfills other functions (e.g., stabilization of protein complexes), because it contains four fatty acid residues, and is a non-bilayer forming phospholipid [38,39]. It is known that cardiolipin is oxidized in mitochondria by X-irradiation [40]. If the ROS are related to cell defense signal transduction, ROS must pass through the membranes and exist in the cytosol to activate signal transduction. In this study, in the mitochondria, we study which ROS can pass through the mitochondrial membrane.

In this paper, we try to clarify which ROS are responsible for signal activation in cytosol through calculations and examination of the literature: •OH, singlet oxygen (^1^O_2_), HO_2_•, •NO, •NO_2_, ONOO^−^, ONOOH. The dipole moments of ROS and RNS are calculated using density functional theory (DFT) calculations. Possible candidates of ROS which pass through the mitochondrial membrane and enter the cytosol to activate the signal transduction pathway are estimated using the calculated dipole moment and experimental permeability coefficient. In addition, the lifetime of each molecule is listed, and ROS that escape from the mitochondria and act as initiators to activate signal transduction in the cytosol are taken into consideration.

## 2. Materials and Methods

### 2.1. Theoretical Calculations of Dipole Moments for ROS and RNS

The dipole moments [41] were calculated according to the dipole information (Table 1). The DFT calculations were performed using Gaussian 09 (Revision A.02, Gaussian, Inc., Wallingford, CT, USA) [42]. The calculations were performed on a 32-processor QuantumCube^TM^ (Parallel Quantum Solutions, Fayetteville, AR, USA) at the B3LYP/6-311++G(3df,3pd) level of theory [43,44,45] with a keyword “polar” to output the molecular polarity (electric dipole moment in D (debye)) [46]. Graphical outputs of the computational results were generated using the GaussView software program (ver. 3.09) developed by Semichem, Inc., Shawnee, KS, USA [47]. The dipole moments were calculated for various ROS and RNS; we calculated the dipole moments of major ROS and RNS that exist in the mitochondria (Table 1): hydroxyl radical (•OH), superoxide (O_2_^•−^), hydroperoxyl radical (HO_2_•), nitric oxide (•NO), nitrogen dioxide (•NO_2_), peroxynitrite (ONOO^−^), and peroxynitrous acid (ONOOH). We also listed the number of molecules of water (H_2_O) and hydrogen peroxide (H_2_O_2_).

### 2.2. Predictive Performance of Mitochondria-Originating Reactive Oxygen Species

The predictive performance of mitochondria-originating reactive oxygen species included the following parameters: the intracellular amount (amount/cell); the half-life; the diffusion distance (µm); permeability coefficients (Pm) (in cm s^−1^); the one-electron reduction potential (*E*^o^) (in V vs. NHE, NHE: normal hydrogen electrode) at pH 7.4; p*K*_a_; and the rate constants for the reaction with ascorbate (AscH^−^) (*k* (AscH^−^)/M^−1^ s^−1^) and glutathione (GSH) (*k* (GSH)/M^−1^ s^−1^) for various ROS and RNS. We focused on ROS generated from the mitochondrial electron transport chain (mtETC). The ROS studied included •OH, singlet oxygen (^1^O_2_), O_2_^•−^, HO_2_•, •NO, •NO_2_, ONOO^−^, ONOOH, alkoxyl radicals (RO•), and peroxyl radicals (ROO•). The H_2_O, oxygen (O_2_), and H_2_O_2_ were also listed. Those radicals were initiated from O_2_^•−^, starting from electron leakage from the ETC and then binding with O_2_. Then, the O_2_^•−^ changed form to become other ROS, such as •OH, singlet oxygen (^1^O_2_), HO_2_•, •NO, •NO_2_, ONOO^−^, and ONOOH, in the mitochondria [2]. •OH and •NO_2_ are constructed by the binding of O_2_^•−^ and •NO. This information was collected from the literature listed in the Table 2 references. To exit the mitochondrial membrane into the cytosol, the ROS should be present in an appropriate amount and have a long half-life, long diffusion distance, large Pm and *E*^o^, and relatively small rate constants for the *k* (AscH^−^)/M^−1^ s^−1^) and *k* (GSH)/M^−1^ s^−1^).

## 3. Results

The results for the calculated dipole moment (in D) and experimental permeability coefficient (in cm s^−1^) are listed in Table 1.

Table 2 shows the predictive performance of the mitochondria-originating ROS. The intracellular amount (amount/cell); the half-life; the diffusion distance (µm); permeability coefficients (Pm; cm s^−1^); *E*^o^, the one-electron reduction potential (V vs. NHE) at pH 7.4; p*K*_a_; and the rate constants for the reaction with ascorbate (AscH^−^) (*k* (AscH^−^)/M^−1^ s^−1^) and glutathione (GSH) (*k* (GSH)/M^−1^ s^−1^) were examined. For considerations of reactions of ROS in the mitochondria, we used AscH^−^ and GSH. Finally, we detected the ONOOH and HO_2_• for the responsible ROS, which crossed the mitochondrial membrane and initiated the intracellular signaling in cytosol (Figure 1).

## 4. Discussion

Majima et al. were the first to report that reactive oxygen species (ROS) generated from the mitochondria promote apoptosis [106], while Itoh et al. described the function of the Nrf2-Keap1 intercellular signal for the first time [107,108]. A recent study described that ROS generated from the mitochondria initiates cellular transduction in the cytosol [34,35]. The further roles of ROS and the subsequent intracellular signals, proteins, and molecule transport change need to be clarified. The establishment of cellular signaling and metabolism change based on mitochondrial ROS augmentation is in demand. Thus, studies on the physiological and pathological functions of mitochondrial ROS will be necessary.

This paper aims to consider the roles of mitochondrial ROS in the activation of intracellular signals. The dipole potential (represented by Ψd) is shown as the potential difference that arises due to the nonrandom orientation of dipolar residues of the lipids and associated water molecules within the membrane [109,110]. ROS with a positive or negative charge cannot escape mitochondria by passive diffusion through phospholipid bilayer due to their large number of dipoles. The results of the dipole moments (Table 1) show that H_2_O_2_ is permeable (the dipole moment is 0.00 D). The dipole moment of •NO_2_ was 0.35 D, indicating permeability. Although the dipole moment of O_2_^•−^ is 0.00 D, the negative charge in O_2_^•−^ precludes its penetration into the membrane. ONOO^−^ is non-permeable. H_2_O (with a dipole moment of 1.89 D), •OH (with a dipole moment of 1.67 D), ONOOH (with a dipole moment of 1.77 D), and HO_2_• (with a dipole moment of 2.23 D) might be permeable. The candidates that can escape from the mitochondria include ROS with small dipole moments, i.e., H_2_O_2_, •NO, •NO_2_, HO_2_•, ONOOH, •OH, and H_2_O. It is well known that •NO_2_ reacts with urate, ascorbate, and GSH at 10^7^ M^−1^ s^−1^ [96]. Therefore, the reaction of •NO_2_ with specific targets in the cytoplasm, where GSH is present at µM~mM levels [111,112], likely occurs with very low frequency [113]. The candidates that can escape from the mitochondria thus include ROS with small dipole moments, i.e., H_2_O_2_, HO_2_•, ONOOH, •OH, and •NO.

The reactivity of ROS/RNS should be essential. However, if the molecules disappear in a short period, there is less chance of the reaction occurring. A greater amount, a long half-life, a greater diffusion distance, a greater Pm, a greater I, a greater one-electron reduction potential, a smaller p*K*_a_, and greater rate constants for the reaction with ascorbate and GSH would be preferable for the studied ROS/RNS. Molecules with electrical charges cannot pass the phospholipid bilayers of mitochondrial membranes [36]. Short-lived molecules, such as •OH, are difficult to contribute to intracellular signaling due to the characteristics of the short-lived molecule (Table 2). For signal activation inside the cytosol, again, H_2_O_2_, HO_2_•, ONOOH, •OH, and •NO can be selected as candidates (Table 2).

It is also essential to consider the conditions that ROS/RNS must overcome to pass through the mitochondrial membrane to become signaling molecules in the cytosol. The plasma membrane consists of both lipids and proteins. The fundamental structure of the membrane is the phospholipid bilayer, which forms a stable barrier between two aqueous compartments. [36]. Most biologically important solutes require protein carriers to cross cell membranes via a process of either passive or active transport. Active transport requires the cell to expend energy to move the materials, while passive transport can be achieved without using cellular energy [37]. Certain substances easily pass through the membrane via passive diffusion, such as O_2_ and CO_2_, along with small relatively hydrophobic molecules, fatty acids, and alcohols [37]. In this study, in the mitochondria, we study which ROS can pass through the mitochondrial membrane.

The ROS produced in the mitochondrial matrix can pass through the two membranes in the mitochondria and enter into the cytosol in order to initiate intracellular signals. Lynch and Fridovich (1978) addressed the question of whether superoxide permeates membranes [114]. The pH of the intermembrane space is lower than that in the matrix due to proton pumping into the intermembrane space; in the intermembrane space (IMS), the concentration of protons is about ten times higher than in the matrix [115]. The pH values obtained were 6.88 ± 0.09 in the IMS, 7.78 ± 0.17 in the matrix, and 7.59 ± 0.01 in the cytosol using a human endothelial cell line, ECV304. [103]. HO_2_• and O_2_^•−^ are of considerable importance in oxidation processes, and the p*K*_a_ of HO_2_•/O_2_^•−^ is 4.8 [62,90]. Therefore, at the physiological pH, HO_2_• hardly exists. In addition to covalent, there is also ionic bonding. There are almost 10 times more protons in the IMS compared to in the matrix. Thus, it may be possible for H^+^ to bind anion molecules, leading to protonation [116]. ROS produced in the mitochondria, HOON- and O_2_^•−^, can be easily protonated in the IMS through ionic bonding. Whereas O_2_^•−^ generated in the mitochondrial matrix may be easily and completely detoxified by mitochondrial SOD, any O_2_^•−^ generated on the outside of the inner membrane will have a longer lifetime and, due to the more acidic environment there than in the matrix, it is likely that O_2_^•−^ will be protonated to HO_2_• and react with a phospholipid in the membrane [117]. Which radicals can penetrate through the mitochondrial membrane? Gus’kova et al. (1984) determined the permeability of the liposomal membrane for O_2_^•−^ and HO_2_•, being P’O_2_^•−^ = (7.6 + 0.3) × 10^−8^ cm s^−1^ and P’HO_2_• = 4.9 × 10^−4^ cm s^−1^, respectively [51]. Cordeiro (2014) described simulations that showed that molecular oxygen (O_2_) accumulated at the interior membrane. Superoxide (O_2_^•−^) radicals and hydrogen peroxide (H_2_O_2_) remained in the aqueous phase and could not enter the membrane. Both hydroxyl (•OH) and hydroperoxyl (HO_2_•) radicals were able to penetrate deep into the lipid headgroup region in the membrane [118]. ROS are produced in the mitochondria, and to establish which ROS can pass through the membrane, we needed to establish the interactions between ROS and the lipid membrane. Cordeiro evaluated HO_2_, O_2_^•−^, •OH, and H_2_O_2_ in terms of the residence times in the phospholipid headgroup region, reported in units of ns [118]. The results show that HO_2_ and O_2_^•−^ have residence times of 17.3 and 12.4 ns, respectively, while •OH and H_2_O_2_ have residence times of 3.8 and 1.5 ns, respectively. A longer residence time suggests a higher affinity for the ROS and phospholipids, and a shorter residence time suggests a lower affinity for the ROS and phospholipids. O_2_^•−^ in the mitochondrial intermembrane space can penetrate the outer membrane mitochondrial membranes through voltage-dependent anion channels (VDACs) [114,119]. However, how much O_2_^•−^ can penetrate through VDACs is unknown. It may be possible for HO_2_• to pass through the membrane without difficulty.

## 5. Conclusions

As a result, HO_2_• and ONOOH were found to be the top candidates to initiate intracellular signaling among the mitochondrial ROS from Table 1 and Table 2. Figure 1 shows the possible ROS that can initiate signal transduction in cells, which are HO_2_• and ONOOH. Further experiments to prove that HO_2_• and ONOOH go out of mitochondria and initiate signals inside cells will be necessary.

## Figures and Tables

**Figure 1 biomolecules-14-00128-f001:**
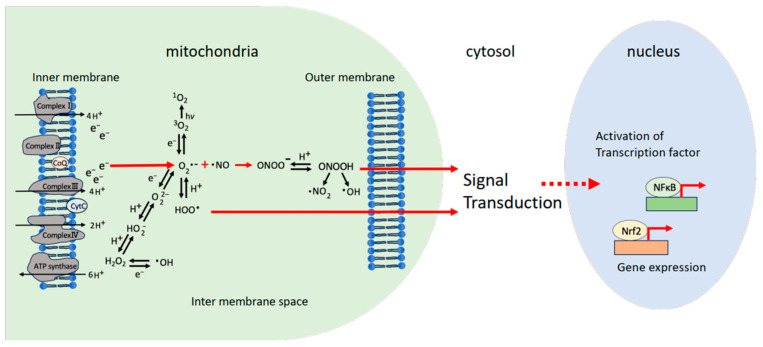
In the mitochondria, 2~3% of electrons leak from the electron transport chain (ETC), and then oxygen traps the electrons, turning them into superoxide anions (O_2_^•−^), and subsequently various ROS are produced: •OH, ^1^O_2_, H_2_O_2_, O_2_^•−^, HO_2_•, •NO, •NO_2_, ONOO^−^, and ONOOH. In the intermembrane space, ten times higher amounts of H^+^ (protons) exist compared to those in the matrix. Among the ROS, ONOO^−^ and HOO• (HO_2_•) can couple with H^+^, and ONOOH and HOO• are produced and penetrate through the membrane, entering the cytosol to initiate intracellular signals, such as NF-κB and Nrf2.

**Table 1 biomolecules-14-00128-t001:** Calculated dipole moment and experimental permeability coefficient of ROS and RNS.

ROS or RNS		Calculated Dipole Moment/D	Permeability Coefficient/cm s^−1^
H_2_O	Water	1.89	2.3 × 10^−3^ [48]
H_2_O_2_	Hydrogen peroxide	0.00 (permeable)	6.1 × 10^−3^, 6.6 × 10^−4^ [49]
•OH	Hydroxyl radical	1.67	
O_2_^•−^	Superoxide	0	1 × 10^−6^ (pH 7.3, 25 °C) [50] (7.6 + 0.3) × 10^−8^ [51]
HO_2_•	Hydroperoxyl radical	2.23	4.9 × 10^−4^ [51]
•NO	Nitric oxide	0.14 (permeable)	93 (20 °C) [52]
•NO_2_	Nitrogen dioxide	0.35	[53] and discussion in the text
ONOO^−^	Peroxynitrite	2.14	Through anion exchanger [54] 8.0 × 10^−4^ [55]
ONOOH	Peroxynitrous acid	1.77	4–13 × 10^−4^ [56,57,58]

**Table 2 biomolecules-14-00128-t002:** Predictive performance of mitochondria-originating reactive oxygen species.

ROS or RNS		Half-Life Time	Amount/Cell	Diffusion Distance (µm)	Permeability Coefficients (P_m_) (cm s^−1^)	*E*^o^′; One-Electron Reduction Potential (V) at pH 7	p*K*_a_	*k* (AscH^–^)/M^−1^ s^−1^	*k* (GSH)/M^−1^ s^−1^
H_2_O	Water	––	––	––	3.3 × 10^−3^ (EYPC) [59]	−2.87 [60] −2.87 [61]	15.7 [62]	––	––
O_2_	Oxygen	––	––	––	12 (DMPC) [59] 125 (DMPC) [59] 114 (DOPC) [59] 157 (POPC) [59] 50 (EYPC: 30% Chol) [59] 38 (RBC human) [59] 21 (CHO cells) [59] 42 (CHO cells) [59]	−0.18 (pH 7) [60] −0.33 [61] −0.16 [63] −0.18 (pH 7, 25 °C) [64]	––	––	––
•OH	Peroxynitrous acid	10^−9^ s [65] 10^−9^~10^−6^ s (diffusion-controlled reactivity) [66] 10^−10^ s [67] 10^−9^ s (1 M, 37 °C) [68] 10^−9^ s [69]	––	3 Å [70] A large flux of hydroxyl radicals would be required to inactivate a substantial fraction of any biological target [70] 0.02 (GSH+) [71]	––	+2.32 (pH 7) [60] +2.31 [61] +2.31 (pH 7, 25 °C) [64] +2.31 [72] +2.31 (pH 7) [73]	11.9 [62] 11.6 [74]	1.1 × 10^10^ (pH 7.4) [61]	1.0 × 10^10^ [72] 1.64 ± 0.01 × 10^10^ [74] 1 × 10^9^ [75] 8.8 × 10^9^ (pH 1.0) [76] 9.0 × 10^9^ (pH 7.6) [77] 1 × 10^10^ [78] 1.1 × 10^10^ (oxidized GSH) [79] 1.4 × 10^10^ (reduced GSH) [79] 1.4 ± 0.1 × 10^10^ (pH 7.8) [80] 4.4 ± 0.5 × 10^10^ (pH 10.6) [80] 2.3 × 10^10^ [81] 4.4 ± 0.5 × 10^10^ (pH 10.6) [82]
^1^O_2_	Singlet oxygen	10^−6^ s [65] 10^−6^ s [67] 10^−6^ s (solvent, 37 °C) [68] 10^−5^ s [69] 10^−9^~10^−6^ s [83]	––	––	––	+0.81 (pH 7, 25 °C) [64]	––	3.2 × 10^8^ [83] 1.8 × 10^8^ [84]	9.39 ± 0.07 × 10^8^ [74]
H_2_O_2_	Hydrogen peroxide	Stable [65] Stable) Stable, decomposed by catalase and GSH peroxidase and by EDTA and ADP [68] Enzymatic [69] 18.1 ± 2.7 min [82]	Physiological condition (proliferation/differentiation/migration/angiogenesis): 0.001~0.1 µM) Stress responses/adaptation (e.g., NRF2): 0.05~5.0 µM [85] Inflammation/fibrogenesis/tumor growth/metastasis: 0.01~10.0 µM [85] Growth arrest/cell death: 1.0~10.0 µM [85]	1600 (GSH+) [78]	6 × 10^−4^ (RBC horse) [59] 3 × 10^−3^ (peroxisome rat liver) [59] 1.2 × 10^−2^ (RBC rat) [59] 2 × 10^−4^ (Jurkat T cells) [59] 3.6 × 10^−4^ (*Chara coralina*) [59] 1.6 × 10^−3^ (*Escherichia coli*) [59] 4 × 10^−4^ (PC12 cells) [59] 1.6 × 10^−3^ (HUVEC cells) [59] 1.1 × 10^−3^ (IMR-90 cells) [59] 4.4 × 10^−4^ (HeLa cells) [59]	+0.39 (pH 7) [60] +0.32 [61] +1.77 [72] +1.8 [78] +0.39 (pH 7, 25 °C) [86]	11.6 [62] 11.75 (pH 7.2) [78]	––	9 × 10^−1^ [72] 9 × 10^−1^ [73] 9 × 10^−1^ (pH 7.4, 37 °C) [78] 8.7 × 10^−1^ [81]
O_2_^•−^	Superoxide	10^−6^ s [65] 1 s (pH 10) [66] 10^−6^ s (diffusion-controlled reactivity) [66] 10^−6^ s [67] The lifetime of superoxide in a cellular environment in water would be expected to be very short, too short to permit diffusion for great distances [68] Enzymatic [69] 3000 ms (10^−6^ M) [87] 175 ms (10^−6^ M + SOD 10^−9^ M) [87] hours (10^−9^ M) [87] 175 ms (10^−9^ M + SOD 10^−9^ M) [87] 0.175 ms (10^−9^ M + SOD 10^−6^ M) [87]	28.4 pM (normal condition)/mitochondria [88] Formation rate (to 6 µM/s) [88] MnSOD-catalyzed dismutation (*k* = 2 × 10^9^ M^−1^ s^−1^) [88] 9.15 × 10^−8^ pmol production/s/mitochondria * 690 nM production/s/mitochondria * 5.5 × 10^4^ superoxide molecules /s/mitochondria *	––	2.1 × 10^−6^ (SBPC) [59] 7.6 × 10^−8^ (EYPC) [59]	+0.94 [72] +0.94 [73]	––	1 × 10^5^ (pH 7.4) [61] 2.7 × 10^5^ (pH 7.4) [61]	~10 to 10^3^ [72] 2 × 10^2^ [81] 1.1 ± 0.04 × 10^3^ [74] 6.7 × 10^5^ (reduced GSH) (pH 7.8) [89]
HO_2_•	Hydroperoxyl radical	51~422 s (pH 2~10) [90] HO_2_• radicals in organic or lipophilic media could have a longer half-life. The half-life of superoxide cannot be calculated unless the concentrations of SOD and all reactive substrates are known [67]	9.15 × 10^−8^ pmol production/s/mitochondria * 690 nM production/s/mitochondria * 5.5 × 10^4^ superoxide molecules/s/mitochondria *	––	4.9 × 10^−4^ (EYPC) [59]	+1.05 (pH 7) [60] +1.06 [72] +1.05 (pH 7, 25 °C) [86]	4 [62] 4.8 [89] 4.8 [90] 4.8 [91]	1 × 10^5^ (pH 7.4) [61] 2.7 × 10^5^ (pH 7.4) [61]	––
•NO	Nitric oxide	ms to s depending on the available concentration of O_2_, otherwise stable [66] Second [67] 1~10 s [69] 445 s [92] •NO:1200 nM in saline: binding with Hb: 2 × 10^5^ M^−1^ s^−1^ [92] Seconds [93]	pM~μM [93] pM~µM in physiological milieu [94] cGMP-mediated processes; <1~30 nM [95] Akt phosphorylation; = 30~100 nM stabilization of HIF-1α; = 100~300 nM [95] phosphorylation of p53; > 400 nM [95] nitrosative stress; 1 μM [95]	––	73 (EYPC) [59] 66 (EYPC: 30% Chol) [59] 18 (RBC human) [59]	–0.52 (pH 7) [60] –0.35 [63] –0.80 [72] –0.80 [73]	––	––	Nondetectable [72] 1.0 × 10^1^ [75]
•NO_2_	Nitrogen dioxide	Second [67] <10 µs [96]	Typically 0.2~0.3 µM [96]	0.4 (GSH+) [78] 0.2 in the cytoplasm [96] <0.8 in blood plasma [96]	~5 (EYPC) [59]	+1.04 (pH 7) [60] +1.04 [63] +1.04 [72] +1.04 [73]	––	1.8 × 10^7^ [96] 3.5 × 10^8^ [96] 3.54 × 10^6^ (pH 5.4~6.5, 55 °C) [97]	3.0 × 10^7^ [72] 2.2 × 10^7^ [75] 3 × 10^7^ [78] 2 × 10^7^ [81] ~2 × 10^7^ [96]
ONOO^−^	Peroxynitrite	0.8 s (pH 7.4) [64] 10^−3^ s [67] 0.05~1 s [69] 0.8 s (pH 7.4) [98] 0.9 s [98] Stable [98] Relatively stable [99] Less than 1 s (pH 7.4, 37 °C) [99] 0.8 s (pH 7.4) [100]	A total peroxynitrite and peroxynitrous acid concentration that exceeds 0.1 mM [101]	60 (GSH+) [78] 0.42 [101]	–––	––	––	7 × 10^2^ [78] 2.35 ± 0.04 × 10^2^, 25 °C [91]	6.6 × 10^2^ (pH 7.4, 25 °C) [71] 7.0 × 10^2^ [73] 6.6 × 10^2^ [75] 1.36 × 10^3^ (pH 7.4, 37 °C) [78] 2.81 × 10^2^ (pH 5.75, 37 °C) [100]
ONOOH	Peroxynitrous acid	Fairly stable [67] 0.90 s, 25 °C [98] Less than 1 s at physiological pH and 37 °C [99] 0.6 s; 1.13 s−1 in phosphate buffer (pH 7.4, 37 °C) [102]	A total peroxynitrite and peroxynitrous acid concentration that exceeds 0.1 mM [101]	––	8 × 10^−4^ (DMPC) [59] 1.3 × 10^−3^ (EYPC) [59] 6.3 × 10^−4^ (DMPC) [59] 4 × 10^−4^ (DPPC) [59]	+1.40 [72]	6.8 [86] 6.8 [90] 6.8 [98] 6.8 [103]	––	6.6 × 10^2^ [72] 1.35 × 10^3^ [81]
RO•	Alkoxyl radicals	10^−6^ s [67] 10^−6^ s (100 mM) [68] 10^−6^ s [69]		––	––	+1.60 [61] +1.60 [72] ~+1.60 [73]	––	1.6 × 10^9^(pH 7.4) [61]	2.76 ± 0.15 × 10^6^ [74]
ROO•	Peroxyl radicals	Seconds to hours depending on conditions [66] 17 s [67] 7 s (100 mM, 37 °C) [68] 7 s [69]		––	––	+1.00 [61] +0.77~1.44 [73] +1.00 [72]	––	1-2 × 10^6^ (pH 7.4) [61]	––

Abbreviations: Chol, cholesterol; DLPC, dilauroylphosphatidylcholine; DMPC, dimyristoylphosphatidylcholine; DOPC, dioleoylphosphatidylcholine; DPPC, dipalmitoylphosphatidylcholine; EYPC, egg yolk phosphatidylcholine; POPC, palmitoyloleoylphosphatidylcholine; RBC, red blood cell. * Assuming a 70 kg man, O_2_ consumption/day is estimated as 14.7 mol/day [14,104]. Assuming that 2% of electrons leak from the mitochondrial electron transport chain (ETC) and that these are trapped by oxygen and made into superoxide, the superoxide production from the ETC is thus estimated as 3402.8 nmol/s. The number of cells/body is estimated as 3.72 × 10^13^ [105]. Thus, superoxide production is calculated as 5.51 × 10^7^ mol/s/mitochondria. Assuming that the volume of mitochondria is 1.32 × 10^7^ m^3^, then considering mitochondrial volume, 1.32 × 10^−16^ m^3^, superoxide production is estimated as 6.90 × 10^2^ µmol/s/m^3^. It is noted that this number is the amount of superoxide produced and that superoxide is modified by other molecules and enzymes, and thus the amount of superoxide existing in cells is much less.

## Data Availability

All data are shown in this paper.

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
