# Peer review of "Mitochondria Play Essential Roles in Intracellular Protection against Oxidative Stress—Which Molecules among the ROS Generated in the Mitochondria Can Escape the Mitochondria and Contribute to Signal Activation in Cytosol?"

_biomolecules, 2024, doi:10.3390/biom14010128_

Round 1
Reviewer 1 Report
Comments and Suggestions for Authors
The authors tried to clarify which ROS or RNS could escape from the mitochondria and activate downstream signals by using calculated dipole moment and 158 experimental permeability coefficient. It was showed that •OH, H2O2, HO2•, NO• and ONOOH were candidates passing through the mitochondrial membrane. Actually, the role of mitochondria in oxidation/antioxidation had been widely addressed. Here, the author might neglected the role of antioxidants in cells, such as GSH. Moreover, authors did not depicted what kind of downstream signal pathways were activated by ROS/RONS released from mitochondria.
In the abstract, the authors mentioned that "Molecules with negative electrical charges never pass through mitochondrial membranes. Short-lived molecules, such as HO•, can never contribute to intracellular signaling." I don't think this is appropriate or scientific description.
In the Key Words, "Reactive oxygen species" was present two times. This was unacceptable.
Comments on the Quality of English LanguageThe quality of English of manuscript is acceptable.
Author Response
The authors tried to clarify which ROS or RNS could escape from the mitochondria and activate downstream signals by using calculated dipole moment and experimental permeability coefficient. It was showed that •OH, H2O2, HO2•, NO• and ONOOH were candidates passing through the mitochondrial membrane. Actually, the role of mitochondria in oxidation/antioxidation had been widely addressed.
Here, the author might neglected the role of antioxidants in cells, such as GSH.
- Thank you for your comments. As you suggested, GSH is probably the most essential molecules in redox status inside cells, including inside mitochondria. In this paper, we focus on "Which Molecules among the ROS Generated in the Mitochondria Can Escape the Mitochondria and Contribute to Signal Activation in Cytosol?", and we listed the ROS in Table 1 and 2. In addition, we listed "the rate constants for the reaction with ascorbate and glutathione are listed for various ROS/RNS: •OH, 1O2, H2O2, O2•–, HO2•, NO•, NO2•, ONOO– and ONOOH.".
In text: Line 120-123: It is written " However, H2O2 from MnSOD could be quickly detoxified by mitochondrial glutathione peroxidase (mtGPx) by reducing it to water [14,18]. This reaction could be accompanied by glutathione, of which the level for most cells is ~5 mM, an excess amount for the reaction [14,18]. ". Please see this explanation.
Moreover, the authors did not depict what kind of downstream signal pathways were activated by ROS/RONS released from mitochondria.
- Line 211-214: We added "Those radicals were initiated from O2•–, starting from electron leakage from ETC and then binding with O2. Then, the O2•– changed form to become other ROS, such as •OH, singlet oxygen (1O2), HO2•, NO•, NO2•, ONOO–, ONOOH, in mitochondria [2]. NO• and NO2• are constructed by binding of O2•– and NO•."
And, I added a discussion for Table 2.
Line 245-261; " This paper aims to consider the roles of mitochondrial ROS in the activation of intracellular signals. The dipole potential (represented by Ψd) is shown as the potential difference that arises due to the nonrandom orientation of dipolar residues of the lipids and associated water molecules within the membrane [51,52]. ROS with a positive or negative charge cannot escape from the mitochondria, due to their large number of dipoles. The results of the dipole moments (Table 1) show that H2O2 is permeable (the dipole moment is 0.00 D). The dipole moment of NO2• was 0.35 D, indicating permeability. Although the dipole moment of O2•– is 0.00 D, the negative charge in O2•– precludes its penetration into the membrane. ONOO– is non-permeable. H2O (with a dipole moment of 1.89 D), •OH (with a dipole moment of 1.67 D), ONOOH (with a dipole moment of 1.77 D), and HO2• (with a dipole moment of 2.23 D) might be permeable. The candidates that can escape from the mitochondria include ROS with small dipole moments, i.e., H2O2, NO•, NO2•, HO2•, ONOOH, •OH, and H2O. It is well-known that NO2• reacts with urate, ascorbate, and GSH at 107 M–1 s–1 [53]. Therefore, the reaction of NO2• with specific targets in the cytoplasm, where GSH is present at µM~mM levels [54,55], likely occurs with very low frequency [56]. The candidates that can escape from the mitochondria thus include ROS with small dipole moments, i.e., H2O2, HO2•, ONOOH, •OH, and NO•."
In the abstract, the authors mentioned that "Molecules with negative electrical charges never pass through mitochondrial membranes. Short-lived molecules, such as HO•, can never contribute to intracellular signaling." I don't think this is appropriate or scientific description.
- Thank you. We changed as
Line 245-250: Molecules with electrical charges are difficult to pass through mitochondrial membranes [36]. Short-lived molecules, such as HO•, can never contribute to intracellular signaling due to the characteristics of the short-lived (Table 1). " Short-lived molecules, such as HO•, are difficult to contribute to intracellular signaling due to the characteristics of the short-lived molecule (Table 2).
In the Key Words, "Reactive oxygen species" was present two times. This was unacceptable.
- Thank you very much. We corrected the keywords.

Reviewer 2 Report
Comments and Suggestions for Authors
The authors address an important issue about the role of mitochondrial generated ROS and RNS that has wide consequences: the transfer from the side of generation in mitochondria and the potential to enter the cytoplasm for initiation of signal transduction cascades. They introduce the possible molecules and select them for theoretical calculations. While they rule out the importance of hydrogen peroxide and superoxide, molecules that are widely considered to be active in signal transduction, they end up with two other molecules, the hydroperoxyl radical and peroxinitrous acid, as top candidates. I find this paper relevant for the field although experimental proof is not provided.
I have a few points that should be addressed by the authors.
1. Please, discuss whether there is experimental evidence available that OH2. and ONOOH play a role in any biological system. If not, how can this proof be provided?
2. Hydrogen peroxide is widely believed to be released by mitochondria and being involved in signaling. The authors use work by Codeiro (2014) that shows that hydrogen peroxide does not enter membranes, and thus remove it from the list of ROS available for signaling in the cytoplasm. Do the parameters in Tab. 1 and 2 exclude H2O2 from being permeable? And how compare the dipole moment and the permeability coefficient of OH2. and ONOOH to that of H2O2. At least in Tab. 1 it is indicated that H2O2 is “permeable”.
3. Is there experimental evidence that OH2. and ONOOH enter bilayer? According to the argumentation for H2O2 this would be important to show as a control.
4. Are there probes / dyes available to detect OH2. and ONOOH in cells? Have these molecules been detected in cells of any biological system? What is the amount they are present in mitochondria?
5. At several positions (e.g., line 61) it is stated that molecules with negative electric charge can NEVER pass through the mitochondrial membranes. On other positions (e.g., line 334) it is mentioned that they can do so (e.g., via ANT, VDAC). For me this are contradictonary statements. Please, correct.
6. Line 148, Cardiolipin (CL) is indicated to be in the inner and outer mitochondrial membrane. Yes, there is some CL in the inner membrane but not the outer membrane. It is not the main lipid that is forming phospholipid bilayer but fulfills other functions (e.g. stabilization of protein complexes). Because it contains four fatty acid residues it is a non-bilayer forming phospholipid. Please, correct/rephrase.
7. For better readability the name of the ROS/RNS should be mentioned in the tables and not only the formula.
8. In the discussion the authors should make clear that the compounds they identified as signaling molecules need to be rigorously be tested for their function.
9. In the abstract the ROS/RNS which came out as signaling molecules have to be named and not only those that were tested.
Comments on the Quality of English LanguageThe English is ok.
Author Response
Reviewer-2
The authors address an important issue about the role of mitochondrial generated ROS and RNS that has wide consequences: the transfer from the side of generation in mitochondria and the potential to enter the cytoplasm for initiation of signal transduction cascades. They introduce the possible molecules and select them for theoretical calculations. While they rule out the importance of hydrogen peroxide and superoxide, molecules that are widely considered to be active in signal transduction, they end up with two other molecules, the hydroperoxyl radical and peroxinitrous acid, as top candidates. I find this paper relevant for the field although experimental proof is not provided.
I have a few points that should be addressed by the authors.
- Please, discuss whether there is experimental evidence available that OH2.and ONOOH play a role in any biological system. If not, how can this proof be provided?
- Thank you for the very important question. As you suggest, it is difficult to prove by experiments. No one even has been challenged to find the ROS as the candidates, due to the lack of a method to detect OH2.and ONOOH at the signaling site. In this paper, we focused on which molecules can pass the mitochondrial membrane.
I realized that I need to explain more about the importance of proving the role of mitochondrial ROS on signaling in the cytosol.
I added Line 141-165
" The importance of this subject, the mitochondrial ROS come out from mitochondria and initiate the signal transduction inside cells, has been hypothesized by many researchers [23–33]. The role of mitochondrial ROS in initiating signal transductions in the cell cytosol has been the subject of discussion [34]. Indo et al. showed that manganese superoxide dismutase (MnSOD) transfection decreases the expression levels of GATA 1, 3, 4, and 5, which are Nuclear factor kappa-light-chain-enhancer of activated B cells (NF-κB) regulating genes [34]. The results showed that MnSOD transfected cells revealed a decrease in expression compared to those in the control. We previously demonstrated proving mtROS causes intracellular signaling, and we published a paper entitled “Evidence of Nrf2/Keap1 Signaling Regulation by Mitochondrial-Generated Oxygen Species in RGK1 cells” in a Special Issue of Biomolecules entitled “The Physiological and Pathological New Function of Mitochondrial ROS and Intraorganellar Cross-Talks" in 2023 (https://www.mdpi.com/journal/biomolecules/special_issues/0XTJ2MAYET) [35]. They transfected MnSOD gene-contained vectors in a gastric mucosal tumorized cell line, RGK1 cells. They examined the expression levels of NF-E2-related factor 2 (Nrf2), Kelch-like ECH-associated protein1 (Keap1), heme oxygenase-1 (HO-1) and 2, MnSOD, glutamate-cysteine ligase (GCL), glutathione S-transferase (GST), and NAD(P)H Quinone oxidoreductase 1 (NQO1), that are all Nrf2-Keap1 regulating gens. The results of immunocytochemistry staining showed a decrease in those expressions in the MnSOD transfected RGK1 cells compared to those in the control. The transfected MnSOD gene should decrease the mitochondrial ROS levels, so after MnSOD transfection, all decreased expression was shown, suggesting mtROS levels control the levels of Nrf2-Keap1 regulating gens. However, the question of which ROS goes out from mitochondria and contributes to intracellular signaling remains unclear."
- Hydrogen peroxide is widely believed to be released by mitochondria and being involved in signaling. The authors use work by Codeiro (2014) that shows that hydrogen peroxide does not enter membranes, and thus remove it from the list of ROS available for signaling in the cytoplasm. Do the parameters in Tab. 1 and 2 exclude H2O2from being permeable? And how compare the dipole moment and the permeability coefficient of OH2. and ONOOH to that of H2O2. At least in Tab. 1 it is indicated that H2O2is “permeable”.
- Thank you. Please see Line 309-317
" Cordeiro (2014) described simulations that showed that molecular oxygen (O2) accumulated at the interior membrane. Superoxide (O2•–) radicals and hydrogen peroxide (H2O2) remained in the aqueous phase and could not enter the membrane. Both hydroxyl (•OH) and hydroperoxyl (HO2•) radicals were able to penetrate deep into the lipid headgroup region in the membrane [64]. ROS are produced in the mitochondria, and to establish which ROS can pass through the membrane, we needed to establish the interactions between ROS and the lipid membrane. Cordeiro evaluated HO2, O2•–, •OH, and H2O2 in terms of the residence times in the phospholipid headgroup region, reported in units of ns [64]. The results show that HO2 and O2•– have residence times of 17.3 and 12.4 ns, respectively, while HO• and H2O2 have residence times of 3.8 and 1.5 ns, respectively. A longer residence time suggests a higher affinity for the ROS and phospholipids, and a shorter residence time suggests a lower affinity for the ROS and phospholipids. It is known that O2•– can penetrate through the anion channel of the membrane [65]. O2•– in the mitochondrial intermembrane space can penetrate the outer membrane mitochondrial membranes through voltage-dependent anion channel (VDAC) [66].
- Is there experimental evidence that OH2.and ONOOH enter bilayer? According to the argumentation for H2O2this would be important to show as a control.
- Thank you. The answer is the same as above A. 2.
- Are there probes / dyes available to detect OH2. and ONOOH in cells? Have these molecules been detected in cells of any biological system? What is the amount they are present in mitochondria?
- I agree with your question. However, I don't have the method to detect OH2. and ONOOH. I think no one knows the method.
- At several positions (e.g., line 61) it is stated that molecules with negative electric charge can NEVER pass through the mitochondrial membranes. On other positions (e.g., line 334) it is mentioned that they can do so (e.g., via ANT, VDAC). For me this are contradictonary statements. Please, correct.
- Thank you. I deleted Line 68-69, and Line 76-77, "Molecules with negative electrical charges are difficult to pass through mitochondrial membranes."
- Line 148, Cardiolipin (CL) is indicated to be in the inner and outer mitochondrial membrane. Yes, there is some CL in the inner membrane but not the outer membrane. It is not the main lipid that is forming phospholipid bilayer but fulfills other functions (e.g. stabilization of protein complexes). Because it contains four fatty acid residues it is a non-bilayer forming phospholipid. Please, correct/rephrase.
- Thank you. We corrected and we added "It is not the main lipid that forms phospholipid bilayer but fulfills other functions (e.g. stabilization of protein complexes), because it contains four fatty acid residues, and is a non-bilayer forming phospholipid."
- For better readability the name of the ROS/RNS should be mentioned in the tables and not only the formula.
- Thank you. We put the ROS's name.
- In the discussion the authors should make clear that the compounds they identified as signaling molecules need to be rigorously be tested for their function.
- Thank you.
I put, 5. Conclusion
As a result, HO2• and ONOOH were found to be the top candidates to initiate intracellular signaling among the mitochondrial ROS from Table 1 and Table 2. Figure 1 shows the possible ROS that can initiate signal transduction in cells, those are HO2• and ONOOH, although further experiments to prove HO2• and ONOOH go out of mitochondria and initiate signals inside cells will be necessary.
- In the abstract the ROS/RNS which came out as signaling molecules have to be named and not only those that were tested.
- Yes, I have corrected the Abstract. Thank you for your comments.

Round 2
Reviewer 2 Report
Comments and Suggestions for Authors
In the revised version of the manuscript, the authors have addressed and corrected most of the concerns related to their first submission. Although experimental proof of their conclusions is missing, I consider this paper relevant for publication, because it makes aware that the importance of specific ROS as it is stated in the literature again and again is not rigorously proofed so far.
Before publication I, however, suggest some more minor changes in oder to make the message clearer.
Abstract, line75: please rephrase sentence to “…negative charges cannot directly diffuse through the phospholipid bilayer of the mitochondrial membranes.”
Line 209: change to “We focused on ROS generated …”
Line 247: Change to “…escape mitochondria by passive diffusion through phospholipid bilayer…”
Line 265: Change to: “… cannot pass the phosholip bilayers of mitochondrial membranes”.
Lines 312-315: Please combine the two sentences to one sentence. As is stands now, they duplicate each other,
.
Author Response
Dear reviewer,
I have changed your suggestions accordingly.
Thank you very much for your valuable suggestions. Please look at biomolecules-new-hjm-8 as attached.
Best regards,
Hideyuki J. Majima